# Prevalence and Correlates of Self-Medication Practices for Prevention and Treatment of COVID-19: A Systematic Review

**DOI:** 10.3390/antibiotics11060808

**Published:** 2022-06-16

**Authors:** Oluwasola Stephen Ayosanmi, Babatunde Yusuf Alli, Oluwatosin Adetolani Akingbule, Adeyemi Hakeem Alaga, Jason Perepelkin, Delbaere Marjorie, Sujit S. Sansgiry, Jeffrey Taylor

**Affiliations:** 1College of Pharmacy and Nutrition, University of Saskatchewan, Saskatoon, SK S7N 5E5, Canada; osa355@usask.ca (O.S.A.); jason.perepelkin@usask.ca (J.P.); delbaere@edwards.usask.ca (D.M.); 2Department of Dentistry, McGill University, Montreal, QC H3A 0G4, Canada; babatunde.alli@mcgill.ca; 3Department of Community Health and Kinesiology, University of Illinois Urbana-Champaign, Urbana, IL 61801, USA; oa03@illinois.edu; 4Grand River Hospital, Kitchener, ON N2G 1G3, Canada; hakeem.alaga@grhosp.on.ca; 5College of Pharmacy, University of Houston, Houston, TX 78712, USA; sansgiry@central.uh.edu

**Keywords:** self-medication, COVID-19, pandemic, home remedies, non-prescription drugs

## Abstract

It has been suggested that the COVID-19 pandemic led to an increase in self-medication practices across the world. Yet, there is no up-to-date synthesized evidence on the prevalence of self-medication that is attributable to the pandemic. This study aimed to conduct a systematic literature review on the prevalence and correlates of self-medication for the prevention and treatment of COVID-19 globally. The review was registered with the PROSPERO database. Searches were conducted following PRISMA guidelines, and relevant articles published between 1 April 2020 and 31 March 2022 were included. Pooled prevalence rate was conducted using the Meta package in R. A total of 14 studies from 14 countries, which represented 15,154 participants, were included. The prevalence of COVID-19-related self-medication ranged from 3.4–96%. The pooled prevalence of self-medication for this purpose was 44.9% (95% CI: 23.8%, 68.1%). Medications reported by studies for self-medication were antibiotics (79%), vitamins (64%), antimalarials (50%), herbal and natural products (50%), analgesics and antipyretics (43%), minerals and supplements (43%), cold and allergy preparations (29%), corticosteroids (14%), and antivirals (7%). The prevalence of self-medication with antibiotics is concerning. More public health education about responsible self-medication amidst the COVID-19 pandemic and future pandemics is required to mitigate the rising threat of antimicrobial resistance.

## 1. Introduction

For over two years, the COVID-19 pandemic has caused significant changes in the way people live. Several measures were introduced across the globe, including face-masking, social distancing, and lockdowns [1,2,3,4]. Importantly, individuals have resorted to self-care measures to mitigate the spread of the virus and to allay their fears. Among these self-care measures is the practice of self-medication, which involves the possession and usage of medicinal products without a prescription from a physician or pharmacist [5].

There has been a surge in self-medication activities since the onset of the ongoing COVID-19 pandemic because of the societal perception of risk and the urge to do something for preventative and curative reasons [6,7,8,9,10,11,12]. Naturally, the scientific response to this surge was for researchers to investigate the prevalence of self-medication, the reasons for self-medication, the common agents for self-medication and the outcomes of this practice because of the COVID-19 pandemic.

Since the onset of the current pandemic, researchers have gathered evidence on medications used by the public to prevent and treat COVID-19 without prescription from health professionals. Although the findings varied across countries, cultures and communities, there are some interceptions in the evidence. The relevance of these studies in developing health education plans and guiding society on safe practices cannot be over-emphasized.

However, there is a need to systematically synthesize the available data on self-medication due to the pandemic to enhance their utilization. A systematic literature review helps with knowledge synthesis and translation, which is relevant to public health education and scientific recommendations. Therefore, we aimed to systematically synthesize available evidence on self-medication practices for the treatment and prevention of COVID-19.

### Objectives

The primary objective of this study was to conduct a systematic literature review and meta-analysis on the prevalence of self-medication practices instituted to prevent and treat COVID-19. Secondarily, the study assessed the factors influencing the self-medication practices instituted to prevent and treat COVID-19.

## 2. Methods

The Preferred Reporting Items for Systematic Reviews and Meta-Analyses (PRISMA) was followed in conducting and reporting this systematic review [13]. A protocol was developed in collaboration with all authors and registered prospectively on the PROSPERO systematic review database (CRD42021248132).

### 2.1. Data Source

Data sources searched include PubMed, OVID-Medline, CINAHL, Embase and Google scholar databases. The search was limited to articles published between 1 April 2020 and 31 March 2022. An additional search was made from citations and references of papers that were considered eligible.

### 2.2. Inclusion and Exclusion Criteria

Only primary, observational peer-reviewed research articles involving adults who provided information about their self-medication practices during the COVID-19 pandemic and were published between April 2020 and March 2022 were included. Prevention and treatment of COVID-19 were considered as one concept. Studies must have assessed self-medication practices instituted to prevent and treat COVID-19 and not merely assessed overall self-medication practices during the period of the pandemic. No language restriction was applied in the selection of articles. A study was included if it reported the practice of self-medication by participants for prevention and treatment of COVID-19 based on the standardized definition of self-medication [14].

### 2.3. Search Strategy

A search strategy was developed with a librarian. Boolean logic was used to combine mesh terms and text words. Truncation of terms was used to increase the inclusiveness in the text word category. In addition, the references of the studies that met the inclusion criteria were searched to identify additional articles for inclusion. The search terms were combined as follows:

(((((((((((((((self medications [MeSH Terms]) OR (nonprescription drugs[MeSH Terms])) OR (alternative medicine[MeSH Terms])) OR (over the counter drugs[MeSH Terms])(self medicat*[Text Word])) OR (nonprescription*[Text Word]))) OR (home care*[Text Word])) OR (alternative med*[Text Word])) OR (over the counter*[Text Word])) AND (coronavirus, sars[MeSH Terms])) OR (pandemics[MeSH Terms])) OR (coronavirus[MeSH Terms])) OR (coronal virus[Text Word])) OR (sarsCOVID[Text Word])) OR (COVID-19[Text Word]).

Added to the above terms to enhance inclusiveness were the searched keywords: (self-medication OR alternative medicine OR home remedies and COVID-19).

### 2.4. Study Selection

Two authors independently applied the search strategies to the search engines and screened them for inclusion. They first screened the search results by title and abstract according to the inclusion criteria by manual screening from the databases. Later, they reviewed the full text of the relevant studies to determine whether they were appropriate for study inclusion. Discrepancies were consulted with another author and resolved by consensus.

Two of the studies reviewed for inclusion eligibility were originally published in Spanish; others were published in English. The Spanish language manuscript was translated to English using Google translator to assess inclusion and enhance easy extraction of the needed information from the studies.

### 2.5. Study Data Management and Extraction

A predefined data extraction Excel spreadsheet developed and approved by all authors was used to extract the relevant data from all the articles included. The data items pulled included the article title, first author, study sample size, study location, study design, the prevalence of self-medication, agents used for self-medication, reasons for self-medication, and statistical correlates of self-medication and the sources of information for self-medication. After the data extraction, all authors reviewed the content of the data extraction form to check for accuracy and determine the suitability of the categorization.

### 2.6. Assessment of the Risk of Bias

To assess the methodological quality of the studies included, the risk of bias of each article was independently evaluated by two reviewers using the AXIS critical appraisal tool specific for cross-sectional studies [15]. Studies were assessed based on clarity of study objectives, appropriateness of study design, justification of sample size, response rate, non-response bias, internal consistency of the results, explanation of the results in the discussion and conclusion, identification of limitations, statement of conflicts of interest/funding and ethical approval of the study. There were 20 items on the checklist. Each reviewer independently rated studies by selecting the appropriate response for each item (“Yes,” “No,” “not described,” “not disclosed,” or “Not stated”). These responses were adapted into numeric scores. All items checked as “Yes” in the checklist were assigned a score of 1, whereas objects in the checklist marked as “No,” “not described,” “not disclosed,” or “not stated” were assigned a score of 0. The total score for each study was then summed up and calculated as a proportion (%) of the total items to indicate the quality of the study. To simplify the interpretation of the AXIS reporting system, studies with ratings of 80% and above were considered good quality, 50–70% as fair quality and less than 50% were considered poor quality.

### 2.7. Statistical Analysis

A qualitative synthesis of the included studies was conducted using descriptive statistics. Meta-analysis of the prevalence rate across the studies was performed using the Meta package (v4.17-0) in R [16]. The pooled prevalence was estimated using the “metaprop” function of the package which allows for logit transformation of proportions and pooling with the generalized linear mixed-effects model, a two-step process that has been shown to be superior to the other available methods for meta-analysis of proportions [17].

## 3. Results

### 3.1. Study Selection

The initial comprehensive search of the databases yielded a total of 1341 publications. During the preliminary screening of the abstracts, 1080 items were left after removing duplicate articles. A total of 720 items were excluded based on the exclusion criteria and relevance. From the remaining 360 items, 325 more items were excluded after reviewing the full reports, leaving 35 articles eligible for further evaluation. After further screening, 21 additional papers were excluded because the studies did not specifically assess self-medication for the prevention and treatment of COVID-19. Therefore, 14 articles met the inclusion criteria used for the current study. Figure 1 displays the PRISMA-based flowchart of the study selection process.

#### 3.1.1. The Methodological Quality of the Included Studies

The AXIS critical appraisal tool specific for cross-sectional studies was used to assess the methodological quality of the included studies (Table 1). Overall, 71.4 percent of the selected studies accrued 16 or more points out of the 20 points in the appraisal tool, indicating a moderate risk of bias. Two studies had 13 points, and two studies accumulated 12 points. All the studies did not provide information about non-responders, nor did they address and categorize the non-responders, thereby raising a possibility of non-response bias in the surveys. Three studies objectively measured the internal consistency of their survey instrument [6,18,19]. One study did not provide information on ethical approval for their survey [12]. Four studies did not justify the sample size [12,20,21,22]. Nine out of fourteen studies used online surveys and convenient sampling methods, which suggests that the findings may not represent the study population, thereby limiting the generalizability of the results.

#### 3.1.2. Studies’ Characteristics

A total of 15,154 participants were pulled together from 14 studies [6,11,12,18,19,20,21,22,23,24,25,26,27,28]. Study characteristics are shown in Table 2. The studies reported self-medication practices for COVID-19 in 14 countries: Australia, Nigeria, Peru, Togo, Jordan, India, Pakistan, Indonesia, Ecuador, Iran, Columbia, Norway, Sweden and the Netherlands. None of the studies came from North America or the United Kingdom. One study reported participation from three European countries. The 14 studies included between 290 and 3792 participants. One study was among the elderly population above 60 years [11]; all studies were among adults, but one defined the adult range as 16–60 years [12]. One study was conducted among women aged 18–49 [19], whereas female participation in the remaining studies ranged from 28.3 percent to 66.3 percent. Where a study indicated self-medication practices for reasons other than COVID-19 prevention and treatment, only participants who used self-medication for a COVID-19-related purpose were extracted.

### 3.2. Prevalence of Self-Medication for Prevention and Treatment of COVID-19

The prevalence of self-medication for prevention and treatment of COVID-19 varied across the individual study population from 3.6 percent to 96.2 percent. The lowest prevalence was from the European study, where it was shown that the prevalence of self-medication for the prevention and treatment of COVID-19 was 3.4 and 0.2, respectively. The Australian study assessed the prevalence of antibiotic use and found a prevalence of 19.2 percent, whereas the highest prevalence was from the Ecuadorian study assessing herbal medicine use. Figure 2 shows that the pooled prevalence of self-medication across the studies for this purpose was 44.9% (95% CI: 23.8%, 68.1%).

### 3.3. The Agents Used for Self-Medication

Antibiotics were reported by 79 percent of the studies as an agent used for self-medication for both prevention and treatment of COVID-19 (Figure 3). 

Azithromycin, penicillin, doxycycline, amoxicillin, ciprofloxacin, erythromycin, metronidazole, levofloxacin and cephalosporins were mostly reported across the studies (Table 3). One study only mentioned antibiotics as agents for self-medication without specifying the name(s) of the antibiotics [20]. Another study said antimalarials (without a specific name) as a separate agent from hydroxychloroquine and chloroquine [18]. We categorized hydroxychloroquine and chloroquine as antimalarials in our descriptive analysis. Some studies did not specify the name of the agents, but provided class names such as herbal products, vitamins, and minerals, whereas a few studies gave the exact name of the agent.

### 3.4. Reasons for Self-Medication and Correlates of Self-Medication

We categorized the reasons for self-medication into the prevention of COVID-19, treatment of COVID-19 symptoms and anxiety about COVID-19. Eighty-six percent of the studies identified the prevention of COVID-19 as the reason for self-medication (Figure 4). Two studies identified fear (anxiety) as the main reason for self-medication; one of these two studies listed the anxiety category as: fear of stigmatization if COVID is contracted, fear of being quarantined, fear of infection, or contact with suspected or known cases [18].

Nine studies described factors that were statistically correlated with self-medication. These include age, gender, education, occupation, knowledge, distrust in health personnel or institutions, fear of being sanctioned or fined for leaving the home, and individual experience [6,18,20,21,22,23,24,25,26] There was no consensus about age as a correlate of self-medication. A study observed that younger age is positively associated with self-medication for preventing COVID-19 [20]. Another study reported being older as a positive correlate of self-medication for preventing COVID-19 symptoms [6].

Four studies reported female gender as positively associated with self-medication [18,21,23,26]. Two studies observed a higher tendency among male genders; however, a higher proportion of the studies were dominated by male respondents, creating a sampling bias [20,22]. Three studies identified working in the medical/health sector as a positive correlation for self-medication [20,21,23]. Two studies reported having more education as positively associated with self-medication. They also observed a lack of knowledge about the medicinal agent as a positive correlate for self-medication behavior [18,20]. One study specifically observed a higher trend of self-medication (81%) among people with post-graduate education [22].

### 3.5. Sources of Information about Self-Medication

Seven studies included sources of information about self-medication in their observation [12,18,19,21,22,24,26]. The sources identified by these studies included family, friends, healthcare professionals, newspapers, the internet, social media, social networks, product brochures, radio and televisions.

## 4. Discussion

This study reviewed self-medication practices during the COVID-19 pandemic. To the best of our knowledge, this is currently the most up-to-date systematic review of the prevalence of self-medication due to the pandemic. This systematic review revealed a worrisome prevalence rate of self-medication with antibiotics for the prevention and treatment of COVID-19 and such practices were mostly in the low- to middle-income countries.

Previous systematic reviews on self-medication before the pandemic reported a similarly high rate of self-medication, especially in low- to middle-income countries [29,30,31]. A pre-pandemic systematic review reported a wide range for most self-medication across the globe [29]. A pooled estimate of the prevalence of self-medication in adolescents during the pre-pandemic era was 50% (95% CI: 31- 68%). Similarly, in our study the prevalence of self-medication practices varied widely between countries, from as low as 3.6 percent in high-income countries to as high as 96.2 percent in low-income countries with an estimated pooled prevalence of self-medication of 44.9 percent (95% CI: 23.8–68.1%). The only caveat is this self-medication was being performed specifically to counter COVID-19. Although there is no frank basis for comparison between the estimated prevalence of self-medication due to COVID-19 and general self-medication before the pandemic, there seems to be not much difference between the self-medication rate pre-pandemic and as a response to the pandemic. The current report is not unexpected considering the associated anxiety about COVID-19, the diversity of information searches about the pandemic and the increased tendency for misinformation [32,33,34].

Antibiotics were the most common agents used for self-medication across all the included studies. Pre-pandemic systematic reviews reported a high prevalence of antibiotic self-medication, especially in low- to middle-income countries [29,30,31,35,36]. The current report also mainly affirms that viewpoint, in that studies from European countries reported a comparatively lower prevalence of antibiotic abuse for the prevention and treatment of COVID-19 than lower-income countries. The only outlier is the study from Australia that reported a relatively high prevalence of 19.2% for antibiotic self-medication due to COVID-19, which might reflect anxieties around the pandemic specific to this population. Importantly, the timing and population setting for each study differed and that suggests a need for a cautious conclusion about the reports. Hence, this current review may stand as baseline evidence to stimulate interest among self-care researchers from high-income countries to re-examine individual antibiotic self-medication practices through and beyond the current pandemic.

The ease of access to these antibiotics without a health professional’s prescription is worrisome. This persistent and frequent access to antibiotics without prescription continues to pose a major threat to the world as antimicrobial resistance (AMR) increases. According to WHO, AMR is a global health and development threat that requires critical multisectoral action as it remains one of the top ten public health threats facing humanity and is mainly driven by the misuse and overuse of antimicrobial agents [37]. The data reviewed agree with the need to emphasize the prevention of unsupervised use of antibiotics as the prevalence continues to rise.

For future pandemics, we recommend that clinicians and pharmacists engage their patients early in discussions relating to what they might be using for infectious disease prevention and guide them appropriately. Since there is divergent information accessible to patients, obtaining reliable information from healthcare providers will be a strong way of preventing misinformation and encouraging responsible self-medication. Since it is very common for antibiotics to be used for conditions such as allergic rhinitis, common cold and other diseases that are of viral origin when they should not be used, we also suggest more public health education about how antibiotics are generally ineffective for infections of viral etiology, such as COVID-19. Such public health campaigns might serve as “low-hanging fruit” with a high chance of success in the fight against antibiotic resistance.

Additionally, many agents used for COVID-19 prevention and treatment were self-prescribed, thereby posing a drug safety risk [20]. This unsupervised use was encouraged by the anxiety surrounding the declaration of COVID-19 as a pandemic and the ensuing lockdown [33]. The rising urge to take responsibility for one’s health to avoid being quarantined might have promoted self-medication without considering the adverse effects. The reasons for self-medicating with the reported agents also seem reasonable. Most studies observed prevention and treatment of COVID-19 symptoms as the rationale for self-medication. The anxiety about contracting the COVID-19 infection and the consequences motivated some people to self-medicate. However, evidence has shown a high tendency of misinformation during the pandemic, with a possibility of cultural and situational differences affecting how people interpret and respond to misinformation [32]. Hence, there is a need for proper education of patients in such a way that responsible self-medication will be encouraged, whereas inappropriate self-medication will be discouraged [38,39,40].

Furthermore, the identified correlates of self-medication for COVID-19 prevention and treatment were like the ones highlighted in previous similar pre-pandemic systematic reviews [29,30,31,35,36], as correlates of younger age, female gender, higher education and working in the healthcare sectors reported in the reviewed studies were similarly reported [29]. In addition, the observed sources of information for self-medication were not different from earlier reports [29,30,35]. Thus, there seems to be no difference between the socioeconomic and behavioral patterns regarding self-medication before and during the COVID-19 pandemics.

Another observation in our systematic review is the need for researchers conducting cross-sectional studies using a survey instrument to be more rigorous in the study design. Many of the reviewed cross-sectional studies failed to assess their survey tool’s internal consistency and validity. The need to assess non-response bias in surveys is equally important to strengthen their quality. We excluded many studies conducted during COVID-19 on self-medication because they did not report specifically if the medications were used to prevent or treat the pandemic. Although every study design is specific to its objective, we advocate for clarity of research purpose and design techniques that can enhance the quality of data synthesis.

A limitation to our study is that most of the articles included did not use a random sampling technique, thereby preventing good representativeness of their sample size. With the high risk of sampling bias and response bias, the result should be carefully interpreted. However, the similarities between our findings and the results from pre-pandemic studies show a recurrent pattern in the prevalence and correlates of self-medication. It also shows that behavioral health education is cardinal in protecting the public from the risk associated with self-medication.

This review has its strength in that it provided synthesized evidence regarding self-medication practices specific to the prevention and treatment of COVID-19, covering 24 months since the onset of this current pandemic. Additionally, various medications and home remedies used for the prevention and treatment of COVID-19 were highlighted to guide healthcare providers during counselling encounters with their patients.

## 5. Conclusions

The reported prevalence of self-medication with antibiotics for the prevention and treatment of COVID-19, especially in developing countries, is concerning. The high rate of antibiotic self-medication across the studies calls for more public health action than whatever measures currently exist to address the threat of AMR. We call for more health education about how an antibiotic is not effective for an infection of viral etiology and firm regulatory measures to mitigate access to antibiotics over the counter. Such public health measures, if successful, will contribute to the reduction of antibiotic resistance. Furthermore, a need to engage in responsible self-medication while trying to overcome the challenge of the pandemic cannot be overemphasized.

## Figures and Tables

**Figure 1 antibiotics-11-00808-f001:**
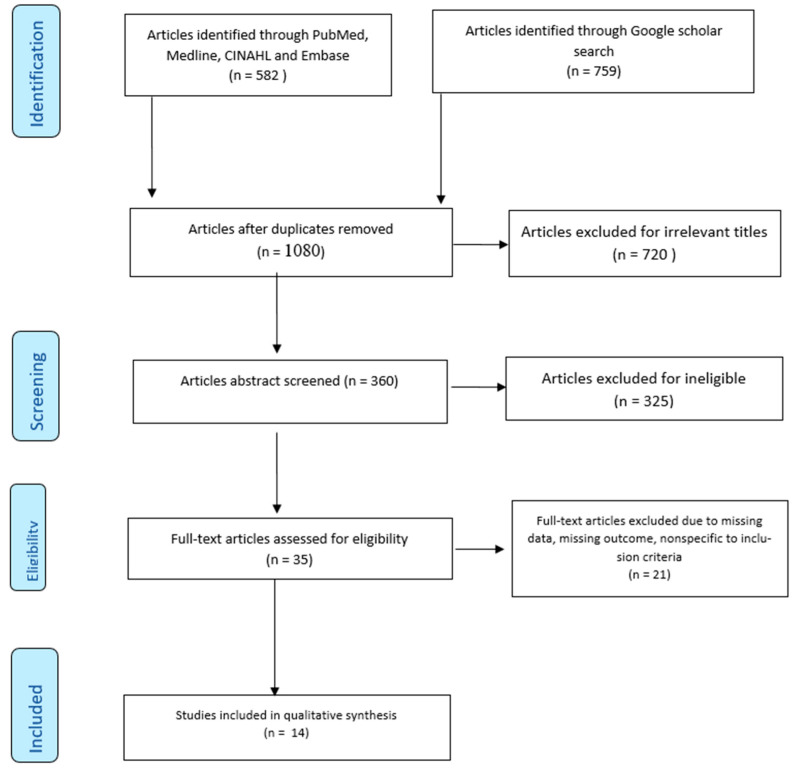
PRISMA flow diagram for the systematic literature review.

**Figure 2 antibiotics-11-00808-f002:**
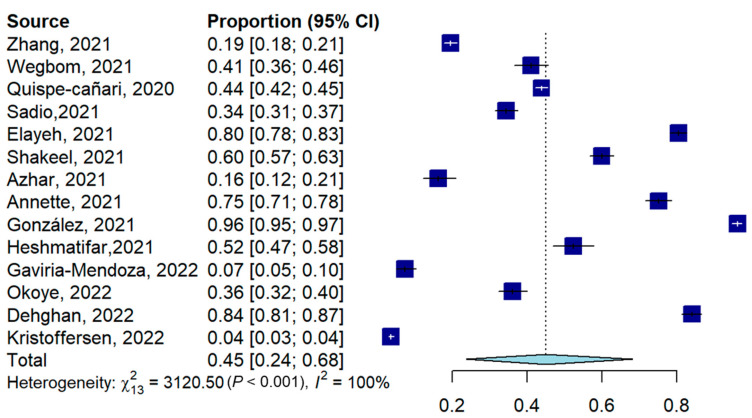
Forest plot for the pooled prevalence of COVID-19-related self-medication [6,11,12,18,19,20,21,22,23,24,25,26,27,28].

**Figure 3 antibiotics-11-00808-f003:**
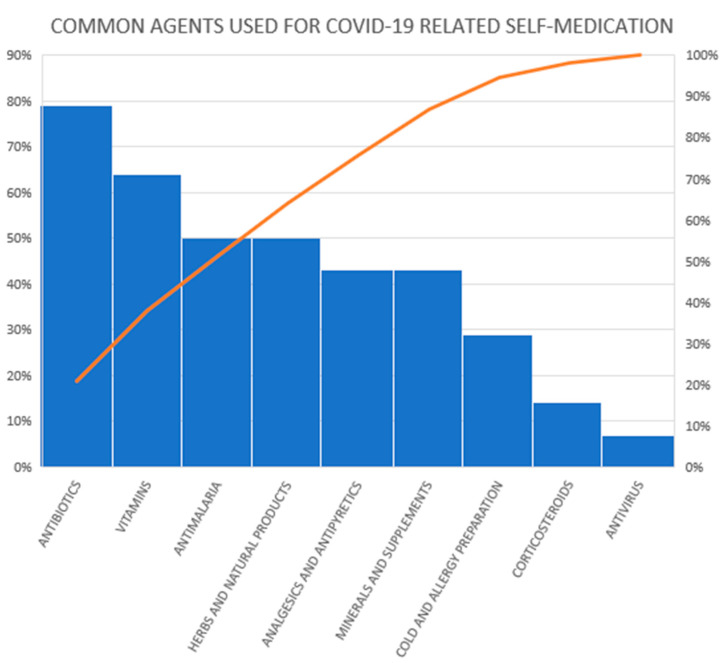
Medicinal agents used for COVID-19-related self-medication in reviewed studies.

**Figure 4 antibiotics-11-00808-f004:**
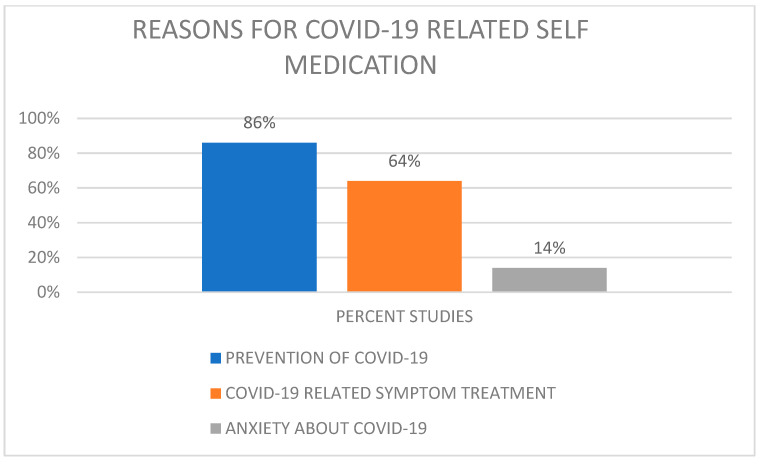
Reasons for self-medication by number of studies.

**Table 1 antibiotics-11-00808-t001:** Assessment of the risk of bias in the reviewed studies.

Assessment Parameters	Zhang, 2021	Wegbom, 2021	Quispe-Cañari, 2020	Sadio, 2021	Elayeh, 2021	Shakeel, 2021	Azhar, 2021	Annette d’arqom, 2021	de los Ángeles, 2020	Heshmatifar, 2021	Gaviria-Mendoza, 2022	Okoye et al., 2022	Dehghan, 2022	Kristoffersen, 2022
**Introduction**														
1. Were the aims/objectives of the study clear?	Yes	Yes	Yes	Yes	Yes	Yes	Yes	Yes	Yes	Yes	Yes	Yes	Yes	Yes
**Methods**														
2. Was the study design appropriate for the stated aim(s)?	Yes	Yes	Yes	Yes	Yes	Yes	Yes	Yes	Yes	Yes	Yes	Yes	Yes	Yes
3. Was the sample size justified?	No	Yes	Yes	Yes	No	No	No	Yes	Yes	Yes	Yes	Yes	Yes	Yes
4. Was the target/reference population clearly defined? (Is it clear who the research was about?)	Yes	Yes	Yes	Yes	Yes	Yes	Yes	Yes	Yes	Yes	Yes	Yes	Yes	Yes
5. Was the sample frame taken from an appropriate population base so that it closely represented the target/reference population under investigation?	Yes	Yes	Yes	Yes	Yes	Yes	Yes	No	Yes	Yes	Yes	Yes	Yes	Yes
6. Was the selection process likely to select subjects/participants that were representative of the target/reference population under investigation?	Yes	No	No	Yes	Yes	No	No	Yes	Yes	No	Yes	Yes	Yes	Yes
7. Were measures undertaken to address and categorize non-responders?	No	No	No	No	No	No	No	No	No	No	No	No	No	No
8. Were the risk factor and outcome variables measured appropriate to the aims of the study?	Yes	Yes	Yes	Yes	Yes	Yes	Yes	Yes	Yes	Yes	Yes	Yes	Yes	Yes
9. Were the risk factor and outcome variables measured correctly using instruments/measurements that had been trialled, piloted or published previously?	Yes	Yes	Yes	Yes	Yes	No	No	Yes	Yes	Yes	Yes	Yes	Yes	Yes
10. Is it clear what was used to determine statistical significance and/or precision estimates? (e.g., *p* values, CIs)	Yes	Yes	Yes	Yes	Yes	ND	Yes	Yes	Yes	ND	Yes	Yes	Yes	Yes
11. Were the methods (including statistical methods) sufficiently described to enable them to be repeated?	Yes	Yes	Yes	Yes	No	Yes	Yes	Yes	Yes	No	Yes	Yes	Yes	Yes
**Result**														
12. Were the basic data adequately described?	Yes	Yes	Yes	Yes	Yes	Yes	Yes	Yes	Yes	Yes	Yes	Yes	Yes	Yes
13. Did the response rate not raise concerns about non-response bias?	Yes	Yes	Yes	Yes	ND	Yes	Yes	Yes	Yes	Yes	Yes	Yes	Yes	Yes
14. If appropriate, was information about non-responders described?	No	No	No	No	No	No	No	No	No	No	No	No	No	No
15. Were the results internally consistent?	ND	Yes	Yes	ND	ND	ND	ND	Yes	ND	ND	ND	ND	ND	ND
16. Were the results for the analyses, as described in the methods, presented?	Yes	Yes	Yes	Yes	Yes	Yes	Yes	Yes	Yes	Yes	Yes	Yes	Yes	Yes
**Discussion**														
17. Were the authors’ discussions and conclusions justified by the results?	Yes	Yes	Yes	Yes	Yes	Yes	Yes	Yes	Yes	Yes	Yes	Yes	Yes	Yes
18. Were the limitations of the study discussed?	Yes	Yes	Yes	Yes	Yes	Yes	No	Yes	Yes	No	Yes	Yes	Yes	Yes
**Others**														
19. Was there information about any funding sources or conflicts of interest that may affect the authors’ interpretation of the results?	Yes	Yes	Yes	Yes	Yes	Yes	Yes	Yes	Yes	Yes	Yes	Yes	Yes	Yes
20. Was ethical approval or consent of participants attained?	Yes	Yes	Yes	Yes	Yes	Yes	ND	Yes	Yes	Yes	Yes	Yes	Yes	Yes
Aggregate risk of bias rating	16/20 (80%)	17/20 (85%)	17/20 (85%)	16/20 (80%)	17/20 (85%)	13/20 (65%)	12/20 (60%)	17/20 (85%)	17/20 (85%)	13/20 (65%)	17/20 (85%)	17/20 (85%)	17/20 (85%)	17/20 (85%)

Abbreviations: ND—not described; NDis—not disclosed; NS—not stated. Aggregate score: all “No”, “ND” and “Ndis” were added up and subtracted from the number of “Yes” to obtain the aggregate risk of bias.

**Table 2 antibiotics-11-00808-t002:** Characteristics of the included studies.

Author (Year)	Location	Study Period	Study Design	Population	Sample size	SD	Prev	Self-Medication Agent	Reasons for Self-Medication	Correlates of Self-Medication	Sources of Information
Zhang 2021	Australia	March–April 2020	Online survey	Adults 18+	2217	49.8% female	19.5%	Antibiotics	Prevention of COVID-19; self-treatment of COVID-19	Younger age; higher education; male gender; healthcare worker; poor knowledge about antibiotics; psychological distress due to the pandemic	Not reported
Wegbom 2021	Nigeria	June–July 2020	Online survey	Adults (age not specified)	461	57.1% female	41.0%	Vitamin C; multivitamins; antimalarials; amoxicillin; ciprofloxacin; herbal products; erythromycin; metronidazole; hydroxychloroquine and chloroquine	Anxiety about COVID-19; prevention of COVID-19; self-treatment of COVID-19	Female gender; Higher education; Poor knowledge about self-medication	Medical personnel; friend
Quispe-cañari 2020	Peru	25 May to 3 June 2020	Online survey	Adults 18+	3792	54.5% female	43.8%	Acetaminophen; azithromycin; ibuprofen; antiretrovirals; hydroxychloroquine; penicillin	Prevention of COVID-19; self-treatment of COVID-19	Older age; employed; living in the rainforest region	Not reported
Sadio 2021	Togo	23 April to 8 May 2020	Survey	Adults 18+; healthcare, air transport, police, road transport, informal sectors	955	28.3% female	34.2%	Vitamin C; traditional medicines; chloroquine/hydroxychloroquine	Prevention of COVID-19; self-treatment of COVID-19	Female gender; healthcare worker; higher education	Not reported
Elayeh 2021	Jordan	26 March to 16 April 2021	Online survey	Adults	1179	46.4% female	80.4%	Antibiotics (azithromycin and doxycycline); analgesics and antipyretics (paracetamol, ibuprofen and diclofenac); minerals (zinc, magnesium and iron salts); vitamins (vitamins C, D and B and multivitamins); herbals and supplements (propolis, omega-3 fatty acids and immune-boosting supplements); antithrombotics (aspirin and enoxaparin); cold and cough preparations; antihistamines; antiseptics; lozenges; nasal solutions (normal saline or sea water); clove oil; menthol rub	Prevention of COVID-19; self-treatment of COVID-19	Female gender; healthcare worker	Newspapers; pharmacist; friends; internet search
Shakeel 2021	India	May 2021	Online survey	Adults	920	28.6% female	59.9%	Paracetamol; azithromycin; expectorants; ivermectin; doxycycline; corticosteroids; hydroxychloroquine	Prevention of COVID-19; self-treatment of COVID-19	Male gender; older age; higher education; government employees	Family; friends; pharmacists/health professionals; newspapers; books/magazines/journals; radio; television; internet
Azhar 2021	Pakistan	2020 (month unspecified)	Online survey	Adults 16–60 years	290	66.3% female	59.5%	Herbal medicines, sana makhi; azithromycin; hydroxychloroquine; ivermectin; Disprin; softener; dexamethasone; cough syrup; Panadol; ibuprofen; levofloxacin; cephalosporins; vitamin C; vitamin D	Prevention of COVID-19; self-treatment of COVID-19	Not reported	Not assessed
Annette 2021	Indonesia	July–December 2020	Online survey	Adults; mothers 18–49 with school-age children	610	100% female	75.0%	Antibiotics; antipyretics; cold medications; antihypertension; blood glucose-lowering agents; supplements, antioxidants (vitamins and minerals); herbs or natural products (ginger and honey)	Prevention of COVID-19; self-treatment of COVID-19	Not reported	Family; friends; social media; news; product brochures
de los Ángeles 2020	Ecuador	2020 (date unspecified)	In-person and online survey	Adults	829	57.8% female	96.2%	Eucalyptus; ginger	Prevention of COVID-19; self-treatment of COVID-19	Not reported	
Heshmatifar 2021	Iran	2020 (date unspecified)	Online survey	Adults; > 60 years	342	55.5% female	56.4%	Analgesics; vitamins and supplements; anticold; sedative; antibiotics; gastrointestinal drugs; cardiac drugs	Prevention of COVID-19; self-treatment of COVID-19	Not reported	Not reported
Gaviria-Mendoza 2022	Columbia	June–September 2020	Survey	Adults	397	58.20% female	7.40%	Chloroquine; hydroxychloroquine;ivermectin;azithromycin	To prevent COVID-19	Distrust in health personnel or institutions;fear of being sanctioned or fined for leaving the home	Social network
Okoye et al. 2022	Nigeria	March–April 2021	Survey	Adults	638	58.60% female	36%	Ivermectin;azithromycin;vitamin C;chloroquine;zinc	To prevent COVID-19 and treat symptoms	Older age;married;pharmacist;higher annual income	Not assessed
Dehghan 2022	Iran	April–August	Survey	Adults	782	66.60% female	84%	Nutritional supplements such as vitamin D, vitamin C, multivitamin, and others, including vitamin B6, vitamin B complex, vitamin E, zinc, calcium, iron, omega-3, and folic acid, or a combination of supplements	To prevent the transmission of COVID-19 or to reduce anxiety caused by the COVID-19 pandemic or both	Female gender; place of residence; COVID-19 Screening	Friends
Kristoffersen 2022	Norway, (*n* = 990), Sweden, (*n* = 500), and the Netherlands, (*n* = 1004)	April–June 2020	Telephone interview and online survey	Adults	2494	49.7% female	Prevention, 3.4% and treatment, 0.2%	Vitamin C (*n* = 3); prayer for own health (*n* = 3); vitamin D (*n* = 2); omega-3, -6 and -9 fatty acids (*n* = 2); relaxation exercise (*n* = 2); unspecified vitamins and minerals (*n* = 2); ginger (*n* = 1); garlic (*n* = 1); ginkgo biloba (*n* = 1); magnesium (*n* = 1); zinc (*n* = 1); breathing exercise (*n* = 1); unspecified herb (*n* = 1)	To prevent and treat COVID-19	Not assessed	Not assessed

Abbreviations: SD—sample distribution; Prev—prevalence.

**Table 3 antibiotics-11-00808-t003:** Categories of medicinal agents used for COVID-19-related self-medication.

Drug Class	Names of Specified Medications in the Studies
Antibiotics	Azithromycin	Penicillin	Doxycycline	Ciprofloxacin	Erythromycin	Metronidazole	Levofloxacin	Cephalosporins
Antimalarials	Chloroquine/hydroxychloroquine	Quinine	Unspecified antimalarials					
Analgesics and antipyretics	Ibuprofen	Diclofenac	Acetaminophen	Aspirin				
Minerals supplements	Calcium	Zinc	Magnesium	Aluminium	Omega-3 fatty acids	Immune boosters		
Cold and allergy preparations	Cough syrups	Lozenges	Nasal solutions	Clove oil	Menthol rub	Expectorants	Unspecified cold and allergy preparations	
Corticosteroids	Dexamethasone	Unspecified corticosteroids						
Antithrombotics	Aspirin	Enoxaparin						
Anthelmintics	Ivermectin							
Antihistamines	Famotidine	Unspecified antihistamine						
Herbs and natural agents	Ginger	Eucalyptus	Unspecified traditional medicine	Unspecified herbal products	Honey	Sana	Makhi	Propolis
Vitamins	Vitamin C	Multivitamins	Vitamin C	Vitamin D				
Antivirals	Antiretrovirals

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
