# Peer review of "Prevalence and Correlates of Self-Medication Practices for Prevention and Treatment of COVID-19: A Systematic Review"

_antibiotics, 2022, doi:10.3390/antibiotics11060808_

Round 1

Reviewer 1 Report

This study provides a systematic literature review on the prevalence and correlates of self-medication for the prevention and treatment of COVID-19 globally. The study raises serious concerns about the widespread use of antibiotics for self-medication, particularly during the first two years of the COVID-19 pandemic. The manuscript is clearly written, comprehensive and relevant to the current issue of the misuse/overuse of antibiotics during the pandemic.

I only have minor comments:

·       -The authors have not described sources of financial or non-financial support for the review, and the role of the funders or sponsors in the review.

·       -The authors have not declared any competing interests.

Author Response

No source of funding or any financial support from any funding agency for this manuscript. The project is self-sponsored by the lead author.

The authors declared no competing interests in any form.

Reviewer 2 Report

Congratulations on the present paper!

Covid situation generated many problems related to medical service access and self-medication. 

I recommend minor revision for minor spelling errors.

Author Response

Thank you

Reviewer 3 Report

The authors conducted a systematic review to synthetize the available evidence on the prevalence of self-medications attributable to the ongoing pandemic. 14 studies were included for a total of more than 15,000 patients distributed in 17 different countries. The main finding of the study is the pooled prevalence of self-medications that is around 45% even though high variability among studies was reported.

The study is well conducted, solid and accurate in methodology. Even though there is a gap of knowledge, the greatest weakness of the study is, in my opinion, the relevance of the topic for the clinical practice. Nevertheless, it is pleasant to read and interesting.

Secondly, the authors states their concern about the high use of antibiotics (79%). I agree on that. However, I would like to know in which extent this data was driven by the use of Azithromycin rather than other antibiotics since its use was at beginning suggested by guidelines for their supposed anti-inflammatory properties against SARS-CoV-2. On the other hand, the use of other antibiotics has no rationale at all except for anxiety of the patient. Maybe it can be a point of reflection for the discussion.

Similarly, I would have grouped some medications such as azithromycin, steroids and Hydroxychloroquine, Ivermectin and antiretrovirals as anti-COVID medications in opposition to all the rest of the drugs to underline the irrationality of their use.

In line 141 write “seven hundred and twenty” in number to be consistent with other parts of the same sentence.

Author Response

The relevance of this topic to clinical practice is that self-medication with antibiotics may be associated with adverse drug reactions. An example is prolonged QTc associated with azithromycin. 

Since this is a review paper where data are based on findings from the 14 studies, the studies did not obtain any data linking antibiotics with the early presumptuous recommendation of Azithromycin for COVID treatment. Also, the concern here is that these antibiotics were used without prescriptions. This concern and a reflection on the possibility of misinformation leading to antibiotics use were addressed in the discussion

The authors did not want to group medications as anti-covid medications as such medicine lack scientific evidence as recommended treatment and none of them were described as an over-the-counter medicine. Hence, they were not expected to be self-medication.

The phrase, seven hundred and twenty items, was written in words because it is not grammatically correct to start a sentence with numbers, so we decided to start the sentence with the phrase. In places where we wrote out the numbers, they did not start a sentence.

Reviewer 4 Report

Very interesting work.

Author Response

Thank you